# Corona Charging of Isotactic-Polypropylene Composites

**DOI:** 10.3390/polym13060942

**Published:** 2021-03-18

**Authors:** Jolanta Kowalonek, Halina Kaczmarek, Bogusław Królikowski, Ewa Klimiec, Marta Chylińska

**Affiliations:** 1Faculty of Chemistry, Nicolaus Copernicus University in Toruń, Gagarina 7, 87-100 Toruń, Poland; halina@umk.pl; 2Łukasiewicz Research Network–Institute for Engineering of Polymer Materials and Dyes, M. Skłodowskiej-Curie 55, 87-100 Toruń, Poland; boguslaw.krolikowski@impib.lukasiewicz.gov.pl; 3Łukasiewicz Research Network–Institute of Microelectronics and Photonics, Kraków Division, Zabłocie 39, 30-701 Kraków, Poland; eklimiec@ite.waw.pl; 4Faculty of Fine Arts, Nicolaus Copernicus University in Toruń, Sienkiewicza 30-32, 87-100 Toruń, Poland; mch@umk.pl

**Keywords:** isotactic-polypropylene, piezoelectric properties, corona discharges, atomic force microscopy

## Abstract

A new approach to obtaining piezoelectric polymeric films based on the isotactic-polypropylene (i-PP) using corona discharge with the energy of 45 W·min/m^2^ was presented. Detailed analyses with Atomic Force Microscopy (AFM) led to the conclusion that the surface quality was the important factor influencing the possibility of charging the i-PP composites, which was necessary to induce the permanent piezoelectric effect. It has been found that the high surface smoothness of the polymer films contributed to improved piezoelectric properties without the need for an additional polymer modification such as orientation, foaming or doping with fillers. The values of the piezoelectric constant, d_33_, of the studied samples were compared to these values for the analogous systems polarized with a constant electric field of 100 V/μm. The milder conditions of the film polarization during the corona discharge process are sufficient to achieve the electrets in i-PP films. The simple and cheap method proposed can be profitable in obtaining flexible electrets in the form of thin films for the production of personal biomedical sensors.

## 1. Introduction

Piezoelectricity is related to the organized structure of dielectric materials containing permanently ordered electric dipoles or uncompensated electric charges. Usually, crystalline substances (both metals and inorganic or organic compounds) consist of atoms, ions, or molecules (dipoles) in which the positive and negative charges are fully balanced, which is related to the symmetrical arrangement of atoms or ions in the crystalline lattice. However, some of these substances become polarized under mechanical pressure as a result of the deformation of the crystal cells and the disturbance of the symmetry in the charge distribution. This generates opposite sign surface charges on adjacent crystal planes, which form an electrical “micro-circuit” responsible for the direct piezoelectric effect. Aside from the direct piezoelectric effect, there may be also so-called the inverse piezoelectric effect, which is when a substance is deformed because of slight atoms movements induced by an applied electrical voltage [1,2,3].

In dielectric materials, which inherently do not exhibit piezoelectricity, this effect can be caused by implementing (“injecting”) an electric charge during the modification process, e.g., by subjecting a sample to a sufficiently high electric field. In the case of semicrystalline polymers containing non-polarized covalent bonds or ionic structures, e.g., polyolefins, it is necessary to provide excess electric charges in such “injection” process. These delivered charges are transferred and trapped in structural defects, irregularities or external contaminants always present in commercial polymers. Furthermore, an orientation process, that is uniaxial or biaxial stretching of the film, causes an enhancement of the macromolecular order—the lamellae form fibrils that contribute to the increase in the crystallinity degree. Expanding inter-lamellar spaces may favor the trapping electric charges [4,5,6].

Piezoelectric materials are used in many branches of modern technologies (e.g., electronics, telecommunication) as electroacoustic transducers, energy harvesters, motion sensors, quartz watches, alarm devices, or elements of scientific instruments (e.g., scanning tunneling microscopy, STM, and atomic force microscopy, AFM). However, new systems with unique properties are still in demand for special applications. Polymers are such promising materials that can be modified accordingly to bring about a piezoelectric effect and retain the beneficial properties of high molecular compounds, such as flexibility, stretchability, ease of forming, availability and non-toxicity. These features allow the polymers to be used not only in electronics or industry but also in medicine as elements in an implanted pacemaker, wearable blood pressure, and human motion sensors, or even in tissue engineering [7,8,9,10,11,12,13].

Moreover, there is still a lack of simple inexpensive piezoelectric components that could be easily accessible and used as personal motion and pressure sensors necessary to monitor the proper functioning of sick people’s organs or sportsmen subjected to extreme efforts. Such piezoelectric materials, if they are intended for direct contact with the human body, must meet high requirements without causing side effects, e.g., allergies. This condition is not met by inorganic piezoelectric materials containing heavy metals [6,7,11].

Our previous works refer to the possibility of obtaining polymeric piezoelectric composites based on filled isotactic-polypropylene (i-PP), which were polarized in a constant electric field [14,15]. However, it turned out that in the case of some heterogeneous polymer composites, a current breakdown occurs during polarization by this method. This event also occurs in the samples subjected to the orientation process.

To avoid such a problem, in this work, i-PP samples were activated by a corona discharge, which is a subtle polarizing factor and acts only on a sample surface without disturbing the internal structure. This process consists of electrical discharges of relatively low power from special electrodes under atmospheric pressure in the presence of air. The energetic electrons, photons, excited atoms, and molecules as well as generated ions create the non-thermal plasma that extends only in a small area around the electrode (much smaller space than the gap between the electrodes). These active individuals are carried in an electric field, reach a sample surface and interact with polymer molecules. The surface layer becomes modified, i.e., polar groups, crosslinks, and even electrets can be formed [16,17,18,19,20]. This method of modification is often used in industry to improve the adhesion of polymer films to other materials to facilitate printing, sticking, or laminating.

The objective of this work was to check whether corona discharge activation is the appropriate method of piezoelectret creation in polymeric materials. Furthermore, the goal was also to compare the piezoelectric properties of the i-PP films and composites containing aluminosilicate fillers activated with the corona discharge with the previously described systems polarized with the electric field of 100 V/μm intensity.

It is known that the polyolefin composite polarization with a high-intensity electric field allows producing stable electrets in which the injected electric charge lasts at least several months [14,15]. Corona discharges can be also a polarizing factor of hydrophobic polymers. However, the surface hydrophilicity of samples achieved by this method is reversible (or partially reversible); thus, the durability of the piezoelectric properties should be tested. From a practical point of view, however, it is important to produce a material exhibiting the permanent piezoelectric effect because such material can be used, for instance, in medicine for biomedical sensor production. Hence, a long-term stability study of piezoelectric properties is essential.

## 2. Materials and Methods

### 2.1. Materials

Isotactic-polypropylene, i-PP (Moplen 456J, Basell Orlen Polyolefins, Płock, Poland) was used as a matrix for aluminosilicate fillers: Sillikolloid P87 (Hoffmann Mineral GmbH, Neuburg, Germany), perlite PEX-02/20 (Mining and Metal Works, Zębiec, Poland), and glass beads MinTron 7™ (Rock Tron, Bristol, UK), which were added in the amount of 5% by weight.

The characteristics of the fillers have been added to this part.

Sillikolloid P87 contained 80% of SiO_2_, 14% of Al_2_O_3_, and less than 1% of Fe_2_O_3_. The particle size was 6 μm—d97 and 15 μm—d50. The filler particles had the shape of round grains and lamellar aggregates [14,21,22,23].

Perlite PEX-02/20 was built of 65–75% of SiO_2_, 10–18% of Al_2_O_3_, 1–5% of Fe_2_O_3_, and a few percent of alkali metal oxides alkaline earth metal oxides. The particle size was less than 20 μm—60%, 20–32 μm—23.8%, 32–40 μm—5.4%, 40–63 μm—9.9%, 63–100 μm—0.6%. The perlite particles looked like crushed bubbles [15,21,22].

Glass beads MinTron 7 ™ consisted of 48–60% of SiO_2_, 20–30% of Al_2_O_3_, 3–7% of Fe_2_O_3_, and 5–9% of alkali oxides The particle size was 20–30 μm—d90 and 5–9 μm—d50. The filler particles had regular spherical shapes [15,21,22,24].

A detailed description of the materials was included in previous works [14,15,21,22,23,24].

### 2.2. Sample Preparation and Polarization

The polymer films with a thickness of about 100 μm were obtained by extrusion at 190–195 °C using Buhler BTSK 20/40D co-rotating twin-screw extruder (Bühler GmbH, Braunschweig, Germany). Then, the films were oriented (at 3:1 ratio) and activated by corona discharge in air atmosphere on the device described below and shown in Figure 1 and Figure 2.

The corona-discharge unit consists of three subassemblies:Generator powered by 3-phase 400/230 V, 50 Hz current, which is a source of energy for electric discharges when activating limited to 2 kW (Institute for Engineering of Polymer Materials and Dyes, Toruń, Poland);A corona-discharge unit where two electrodes (Institute for Engineering of Polymer Materials and Dyes, Toruń, Poland) are mounted (in parallel to each other);High tension transformer for transferring energy from the generator to the discharge unit (Institute for Engineering of Polymer Materials and Dyes, Toruń, Poland).

The electrode that has been used for corona discharge activation is made of an aluminum profile, of special multi-edged construction (Figure 2). It works together with the insulated supporting electrode (roller electrode). The distance between the activating electrode and the film does not exceed 1–2 mm, the average temperature is in the range of 30 to 35 °C. The unit housing is pre-insulated to avoid electromagnetic disturbance and electric shocks. The provided energy (E = 45 W·min/m^2^, i.e., 27 kWh/m^2^) has been calculated from the relationship:E = M/(V·B)(1)
where M is activator power (36 kW), V—is a film speed (2 m/min) and B is an activated film surface (0.4 m^2^).

The apparatus presented in Figure 1 was designed and constructed to modify polymer films in the process of orientation and corona discharges in one technological line. The parameters of the device allow for the production of several meters long ribbons of a modified polymer 20 cm wide.

### 2.3. Determination of Piezoelectric Properties

The piezoelectric charge was measured systematically over 7 months at stress values up to 120 kPa using contact electrodes and a measuring system consisting of an electromagnetic actuator, Tektronix AWG420 arbitrary waveform generator (Electronic Test Equipment, Cary, NC, USA), Meratronik P334 power amplifier (Meratronik, Warsaw, Poland), XFL212R tensometric force sensor (Measurement Specialties Inc., Hampton, VA, USA), ADR 154 amplifier (FGP Sensors Inc., Clayes Sous Bois, France), Keithley electrometer (Chicago IL, USA), and LeCroy LT-341 oscilloscope (LeCroy, Chestnut Ridge, NY, USA). The obtained values are the average of at least 6 measurements for the same sample.

The same method and conditions for testing the piezoelectric properties of samples were applied for the composites polarized with corona discharges and with an electric field. Maintaining the same measurement methodology is necessary to compare the properties of piezoelectrets created in two different ways.

### 2.4. Atomic Force Microscopy (AFM)

AFM images in a dynamic force mode (tapping) were obtained with a MultiMode Nanoscope IIIa (Veeco Metrology Inc., Santa Barbara, CA, USA) using silicon nitride tips. 2D, 3D images, histograms, and cross-sections of surface images were recorded. Roughness parameters [25]: arithmetic (R_a_) and geometric (R_q_) mean of the surface roughness profile are expressed as:(2)$$R_{a}=\;\frac{1}{N}\overset{N}{\underset{j=1}{\sum }}\left|Z_{j}\right|$$
(3)$$R_{q}=\;\sqrt{\frac{\sum Z_{j}^{2}}{N}}$$
where: Z_j_ is the deviation of a given profile point from the straight line, N is the number of measurement points. R_a_ and R_q_ were calculated for scanning area of 100 µm^2^.

The AFM technique was chosen to study subtle changes in the surface topography of samples on a nanometric scale at high resolution, in particular, to observe the roughness profiles.

## 3. Results

### 3.1. Preparation and Polarization of Polypropylene Films

Manufacturing conditions were appropriately adjusted to produce i-PP films and the composites having each time the same good mechanical properties and macroscopically smooth surfaces [14,15]. The i-PP composites contained fillers composed of silicon oxide (which is a dominant ingredient in the amount of 60 to 80%), aluminum oxide (10–30%), and oxides of Mg, Ca, Na, K, Fe (minor components) [23,24]. Fillers are generally added to polyolefins to reduce production costs, but they also have a significant impact on the product properties, namely, they modify mechanical and processing properties as well as improve thermal stability [14,15]. The i-PP composites form heterogeneous systems due to the different physical properties of the aluminosilicates and the polymer.

The produced polymer films were modified to obtain a surface electric charge with the help of corona discharges using the device described in the experimental part. The device operated in air atmosphere and delivered 45 W·min/m^2^ energy to films. The corona discharges were induced by the ionization of gas molecules near the electrode from which the electric current flows to the second electrode. This is accompanied by the emission of electromagnetic radiation observed as a glow (called the corona). This phenomenon occurs when the strength of the electric field exceeds a certain limit value above which an electric current flows to the second electrode. In the immediate vicinity of the discharge electrode, there are a strong electric field and electromagnetic radiation that excite and ionize air molecules, mainly oxygen (Figure 3). During corona discharges, individuals characteristics of plasma are formed, i.e., positive and negative ions, electrons, excited molecules (e.g., singlet oxygen), free radicals (e.g., oxygen atoms), ion radicals (e.g., superoxide radical ion O_2_^•^). These active particles are accelerated in the electric field and reach the surface of a polymer film placed on the second electrode. The active individuals interact with macromolecules causing surface polarization of the sample, which results in the formation of a piezoelectric material [16,17,18,19,26]. The corona discharges also increase the surface polarity, which was confirmed by contact angle measurements [21,27,28].

The original samples that were not subjected to the corona discharge did not show permanent polarization, and hence piezoelectric effect.

### 3.2. Piezoelectric Properties

To determine the piezoelectric properties of the tested samples, both the piezoelectric charge (q) and the voltage generated as a result of the compressive stress (P) were measured. Both parameters showed the same trend; therefore, only the charge values were demonstrated. The piezoelectric charge coefficient (d_33_) was calculated by dividing the charge by the stress (Equation (4)). The subscript “33” in d_33_ means that the direction of an electric field was parallel to the deformation direction.
d_33_ = q/P (pC/N)(4)

As a result of activation with the corona discharge, the samples of i-PP and its composites with 5% of the filler achieved the piezoelectric charge in the range from 10 to 150 pC/cm^2^. This charge was stable during the first few days in most samples and it was practically constant up to over 200 days of storage at room conditions (Figure 4).

The highest charge density value was obtained for the neat unoriented i-PP. After 200 days, it was 150 pC/cm^2^ (at a load of 100 kPa). A similar systematic decrease in the absolute q value was characteristic of the composite with perlite, while the samples with other fillers were characterized by lower |q| values. These differences can result from various morphology of the fillers, i.e., particle size, shape, and degree of agglomeration, which significantly affects the composite properties [23,24].

Positive q values observed only on the neat oriented i-PP film indicate a different polarization mechanism in this sample, which is still not fully explained [1]. It can be assumed that a positive surface charge after corona discharges on the corrugated surface could result from the reactions of protruding fragments of macromolecules with free radicals formed as a result of interactions of mainly electrons with molecules from the air.

Figure 5 presents the dependences of |q| and d_33_ on stress (P) for the selected samples. The relations are non-linear; therefore, d_33_ values were determined at 100 kPa to compare the piezoelectric properties of the polymer sample with different filler. Moreover, the absolute value of the charge measured was used for the calculations of d_33_ values.

The decrease in d_33_ coefficient at higher stresses is typical of piezoelectric materials, particularly piezopolymers [6,14,15]. The acting force causes the deformation of a material, which affects electrical properties. Initially, at low stresses, the deformation is relatively large due to the presence of free volume between the polymer chains. The increase in the compressive force causes less deformation because the sample structure becomes more compact. More charges are accumulated in a material with a loose structure when compared with a compressed sample where charge accumulation becomes difficult owing to the smaller gaps.

Table 1 contains d_33_ values for the studied samples. These values were higher for the unoriented films compared to the oriented ones. For the unoriented i-PP film, the d_33_ value was 15 pC/N, whereas for the oriented sample, it was 7.6 pC/N. Moreover, the composites were characterized by lower d_33_ values compared to the neat i-PP films, which indicated that the generation of the piezoelectric effect in the presence of the fillers was more difficult. In the case of i-PP with glass beads and Sillikolloid P87, the piezoelectric charge and d_33_ were smaller than these for the films with perlite.

A piezoelectric material can be produced from a nonpolar polymer such as PP when the polymeric material contains fillers or empty cavities (i.e., cellular structure), which can accumulate the supplied electric charge [6,14,15,29,30,31,32,33,34,35]. However, in the case of the film that has no cellular structure or fillers, the formation of stable electrets is possible owing to the presence of some internal irregularities in the polymer chains. Since the content of these irregularities or structural defects (e.g., oxygen-containing groups that may arise during air polymerization or processing) is small, their contribution to the polarization appears to be negligible, but it cannot be neglected. The charge formation at a polymer/air interface occurs when active chemical species (ions, radicals, excited molecules) from corona discharges reach the film surface.

The d_33_ values obtained for the corona discharge activated i-PP composites can be compared with this parameter for poly(vinylidene fluoride), PVDF, which is known as a piezoelectric polymer and can be used as a reference (Table 1 [6,10,36,37,38,39,40,41]). However, it should be mentioned that PVDF is more expensive than i-PP and needs a high electric field for poling, which limits its application [36].

### 3.3. Surface Morphology of Polypropylene Films

The surface morphology of the samples determines their surface properties. The quality of the surfaces was accurately characterized in nanoscale. Besides AFM images, the corresponding depth histograms and cross-sections of surfaces have been analyzed (Figure 6 and Figure 7).

As can be seen, no histogram has a normal (regular) data dispersion, and the shapes of the curves are more or less asymmetrical, which indicates the coexistence of hills and wells of different sizes irregularly distributed on the sample surfaces.

The narrowest depth histograms, centered below 100 nm or 200 nm of height, were recorded for the unoriented i-PP film and the composite with perlite. Additionally, the surface cross-sections of these samples showed the smallest fluctuations in the distance from the average surface (of the order of 30–40 nm). The oriented i-PP film exhibited a much broader depth distribution pattern with long tails and the distance between the extreme point of the elevation and cavities about 150 nm (Figure 6).

The curves of the remaining samples also showed very wide peaks, indicating a large distribution of protruding points on the surface (Figure 7). The highest maximum on the histogram of the composite with MinTron 7 ™ corresponded to about 300 nm. In the composites with the fillers, the film orientation also contributed to a significant increase in surface roughness. The surfaces of the oriented composite films were similar to that for the oriented i-PP, so the composite film images are not shown. Numerous nodules and depressions were observed on the film surfaces although the polymer was plasticized during the extrusion process at 190 °C and the filler particles were well dispersed in the polymer matrix (heterogeneities were invisible to the naked eye).

The relatively smooth surface of the i-PP film with perlite resulted from the morphology of this filler containing a lot of small, dusty fractions (90% particles had diameter lower than 0.1 μm [15] that accurately could fill the cavities formed during the extrusion process. In contrast, both Sillikolloid P87 [23] and MinTron 7 ™ [24] were composed of larger particles: only 50% < 1.5 μm and 50% < 5–9 μm, respectively. Moreover, perlite and MinTron 7™ were completely amorphous, in contrast to Sillikolloid P87, which contained a crystalline phase, as previously confirmed with XRD [21].

The highest value of piezoelectric coefficient d_33_, found for the unmodified and unoriented i-PP film, could be explained by the optimal surface roughness, which was lower than that for the composites containing fillers. Both the addition of the filler to i-PP and the orientation process led to a decrease in the values of the piezoelectric charge and d_33_, while the surface roughness increased. As can be seen, the neat i-PP film exhibited a smoother and flatter surface, where roughness parameters R_a_ and R_q_ were 15.8 nm and 20.5 nm, respectively (Figure 6). The orientation process caused an increase in the surface roughness for the unfilled i-PP film (R_a_ = 36.7 nm, R_q_ = 47.9 nm). In this case, the increase in the surface roughness was associated with the formation of folds due to the stretching of the film in one direction.

The introduced fillers caused the formation of numerous bulges on the surfaces resulting from the heterogeneity of the samples. R_a_ and R_q_ values increased several times in the filled samples.

The corona discharges did not affect the sample morphology significantly, probably because the energy revealed during the corona treatment was too small to induce noticeable changes.

## 4. Discussion

The formation of electrets in nonpolar i-PP is associated with its semicrystalline structure. Isotactic-PP can occur in three crystallographic structures, α-monoclinic, β-hexagonal and γ-orthorombic, but, as our earlier XRD studies showed [15], t monoclinic α-crystals dominate in the used polymer, and the crystallinity degree of the specimens obtained in the extrusion process was about 70%. The macromolecular crystallites, usually in the form of folded lamellas, are embedded in the disordered phase [42]. There are places of heterogeneity and structural defects that are responsible for the accumulation of electric charges supplied in the corona discharge process, which leads to the formation of domains with permanent polarization.

Furthermore, in the case of the composites with the fillers of identical composition, polarized in the constant electric field, the piezoelectric effect was much greater compared to that for the corona discharge activated i-PP composites [14,15]. Therefore, it is worth remembering that the high poling electric field causes permanent polarization in the whole sample volume—the charge is injected into polymeric films and accumulates in the heterogeneous places which are a result of the presence of inorganic fillers. In contrast, activation by means of corona discharge covers only a sample surface and provides the charge needed for the piezoelectric effect only in the case of good contact of the flat film with active species emitting from the electrodes. Therefore, the corona discharge is a mild polarizing factor that does not affect the internal film structure and the piezoelectric effect induced is weaker but enough for practical uses.

Thus, it can be concluded that film smooth surfaces favored obtaining permanent polarization caused by corona discharge because interactions of i-PP macromolecules with the active individuals arising during discharges were facilitated in this case. The significant electric charge accumulated on the smooth sample surface was schematically shown in Figure 3a. On the other hand, on very rough surfaces, the active particles could polarize only the protruding hillocks and probably did not reach more hidden places, which was not enough to create electrets. Thus, the accumulated charges in few points of protruding polymeric chains, above the surface, (Figure 3b) were not enough to obtain the material with good piezoelectric properties.

It is worth mentioning that the tested samples showed good mechanical properties (Young’s modulus determined in the static tensile test was higher than 1000 MPa). Moreover, the orientation process also improved the mechanical properties of the studied samples [14,15] and the corona discharges did not affect them much.

## 5. Conclusions

Corona discharge can be used for the creation of piezoelectrets in cast extruded polypropylene films. The presence of aluminosilicate fillers in the films makes the piezoelectric effect weaker, which is probably due to the effective scattering or deactivation of electrons/ions/particles formed during corona discharges on rough surfaces. It was confirmed by the detailed analysis of the surface profiles and images by AFM.

Although the piezoelectric coefficient, d_33_, reaches moderate values for the samples treated with corona discharges compared to d_33_ values for the samples charged in the constant electric field [15], but it is sufficient for the production of the piezoelectric components for personal use simple devices. The same d_33_ value was recently reported for PP fiber-reinforced lead zirconate titanate [43].

This simple preparation procedure does not require complicated apparatus for the generation of the electric charges in non-polar polymers. There is no need to add mineral fillers to a polymeric matrix in this case, which is associated with one technological operation less in the production process. Moreover, in this case, one can activate very large surfaces quickly (film of any length and width depending on the geometry of the device) as opposed to small sample sizes obtained in the poling in the high electric field (depending on the electrode sizes). Moreover, the unfavorable electric breakdown effect that can occur when the polymeric film is activated by a constant electric field is eliminated.

The electrets made of polymer films of different compositions and morphology have different lifetimes. In the case of corona charging, the lifetime mainly depends on the number and energy of the charge traps on the film surface. Various defects in the chemical structure (e.g., branching of polymer chains or impurities from the polymerization process) and crystalline irregularities (e.g., edges, dislocations) are such traps.

It should be remembered that corona discharges can result in the formation of oxidized polymer structures (hydroxyl and carbonyl groups), which, however, are difficult to monitor with ATR-FTIR spectroscopy, probably because these polar products can be placed in a thinner layer than could be detected by this method. The presence of these groups formed under the influence of corona discharges was confirmed by measurements of contact angles [21,27,28]. Such oxidized groups increase the polarity of the inherently hydrophobic polyolefins, which contributes to the increase in interactions with the charges induced in the activation process.

The piezoelectric charge could be permanently trapped in a specific place but, before it happens, the charge could move to the inner layers of the film due to the semi-crystalline structure of the polymer, which was usually described as a hopping mechanism [44,45]. In this way, the charge may be prevented from neutralization under the influence of external environmental factors, which results in the stabilization of piezoelectric properties.

Furthermore, the piezoelectrets created have the advantages of polyolefins, such as flexibility, non-toxicity, photo- and thermal stability, resistance to external factors, the possibility of forming any shape using available processing methods, and the production of cheap raw material. A short processing and a rapid modification with the corona discharge method also means saving energy. Moreover, the undoubted advantage of the produced i-PP electrets is the stability of the obtained charge in the samples over at least 7 months maintain also their good mechanical properties. These piezomaterials can be an alternative to cellular polymers, which, due to their porosity, are excluded in the environment containing moisture or other penetrating agents that have an adverse effect on piezoelectric properties; for instance, polyolefins [46].

## Figures and Tables

**Figure 1 polymers-13-00942-f001:**
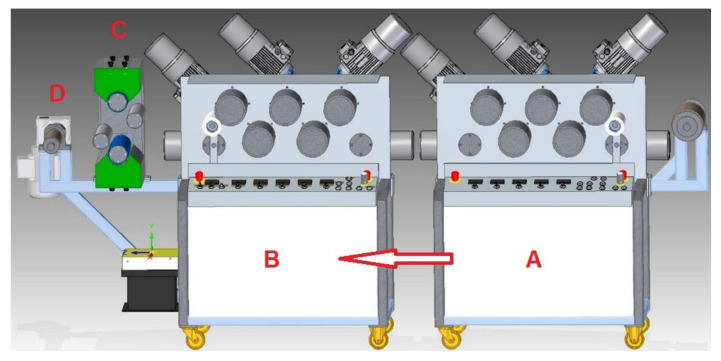
Diagram of the device for film orientation and activation by the corona discharge: A—heating module, B—cooling module, C—activator, D—film winding spool; arrow shows the direction of film transfer.

**Figure 2 polymers-13-00942-f002:**
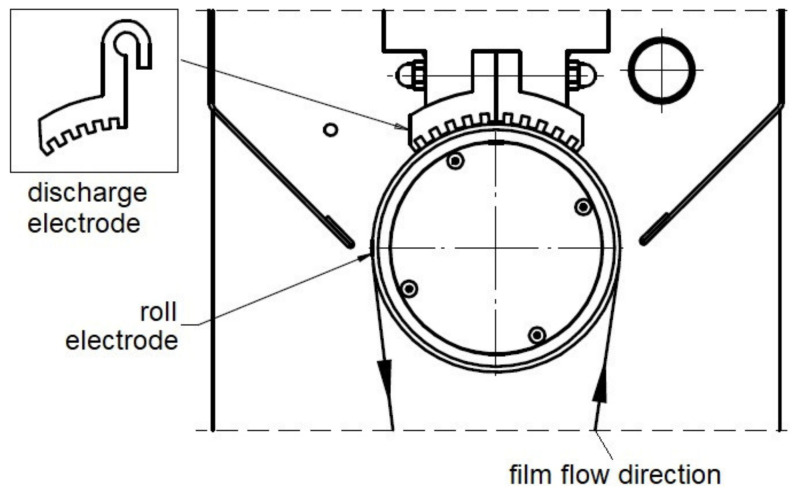
The main part of the corona discharge device (marked as C in Figure 1) coupled to the orientation unit; on the left side at the top—the real shape of the charging electrode.

**Figure 3 polymers-13-00942-f003:**
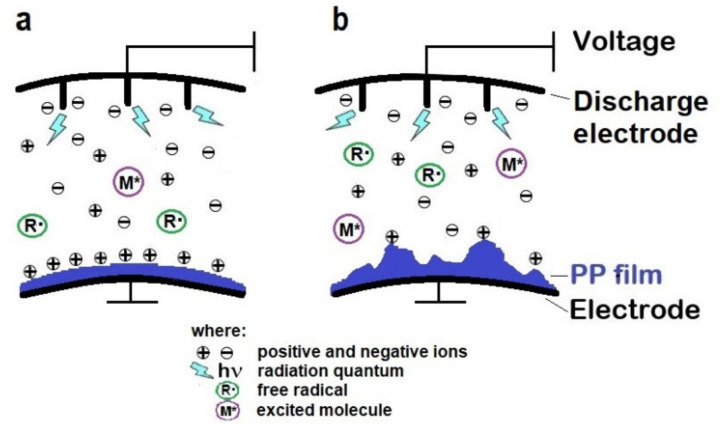
The scheme of the i-PP film polarization process during corona discharges: (**a**)—a smooth film, (**b**)—a rough film.

**Figure 4 polymers-13-00942-f004:**
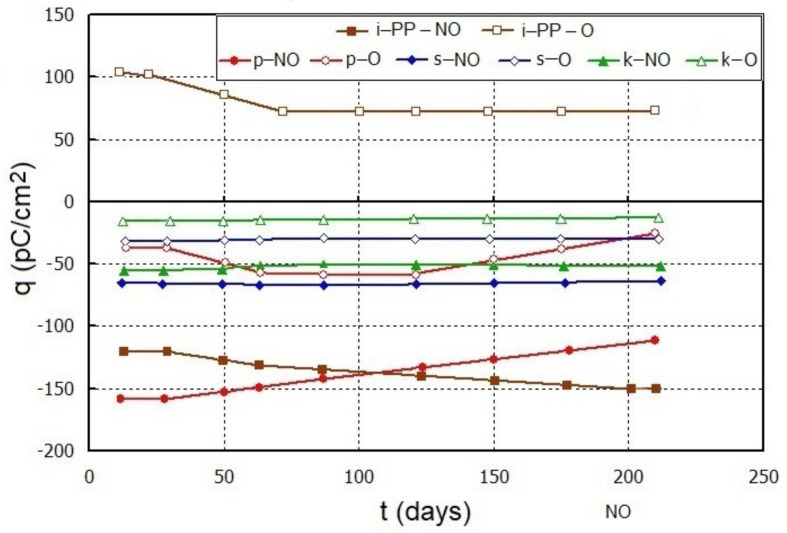
Dependence of the piezoelectric charge of the corona discharge activated i-PP films and its composites with Sillikolloid P87 (s), perlite (p), and MinTron 7™ (k) on storage time (t, days) at room conditions. The piezoelectric charge was determined at the stress of 100 kPa (NO—unoriented, O—oriented films).

**Figure 5 polymers-13-00942-f005:**
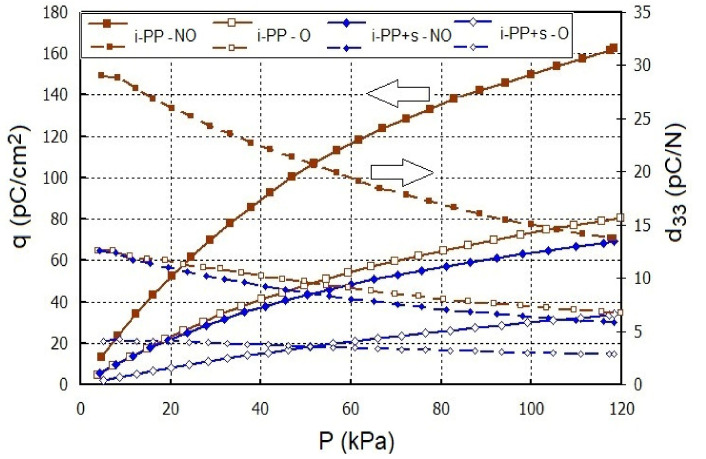
Dependence of absolute values of the piezoelectric charge |q| and d_33_ coefficient on stress for the corona discharge activated i-PP film and the composites with 5% Sillikolloid P87 (s) (NO—unoriented, O—oriented films).

**Figure 6 polymers-13-00942-f006:**
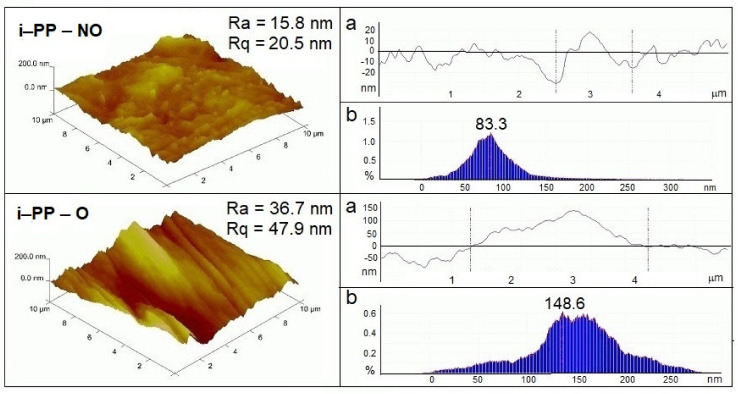
AFM images of the unoriented (NO) and oriented (O) i-PP films. On the left—3D images, on the right—cross-sections of surfaces (**a**) and depth histograms (**b**).

**Figure 7 polymers-13-00942-f007:**
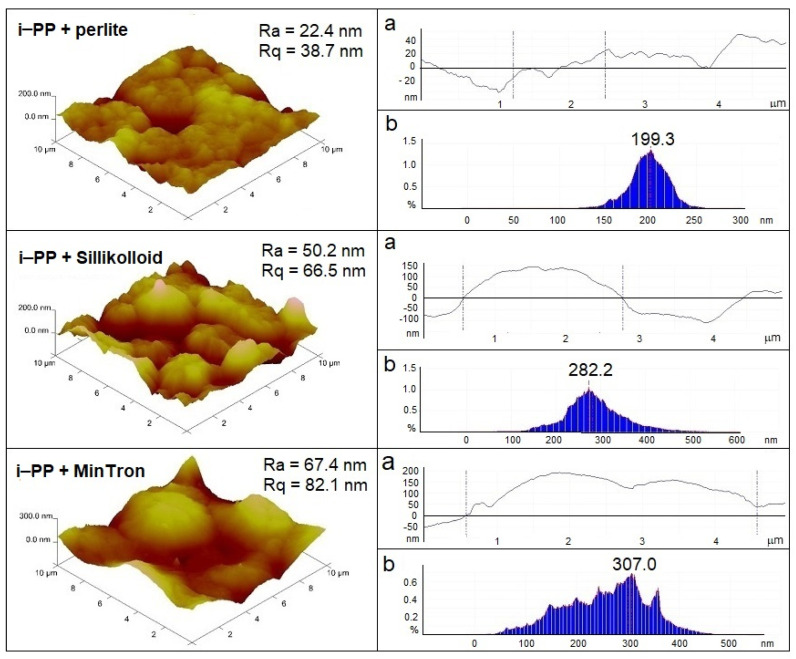
AFM images of the unoriented i-PP composites with 5% of the fillers. On the left—3D images, on the right—cross-sections of the surfaces (**a**) and depth histograms (**b**).

**Table 1 polymers-13-00942-t001:** Piezoelectric coefficient (d_33_) of the corona discharge activated i-PP film and its composites (unoriented and oriented) with 5% of mineral fillers at 100 kPa stress. The d_33_ values in brackets were obtained for the samples of the same compositions prepared under the same conditions but activated with a constant electric field of 100 V/μm (values taken from references [14,15]). d_33_ values for PVDF are provided from the literature for comparison.

Sample	d_33_ (pC/N)	
	Unoriented	Oriented
i-PP	15 (14)	7.6 (18)
i-PP + perlite PEX-02/20	9.6 (33)	2.5 (80)
i-PP + Sillikolloid P87	7.0 (52)	3.0 (-) ^1^
i-PP + MinTron 7 ™	4.9 (78)	4.0 (-) ^1^
	d_33_ (pC/N)	Ref.
PVDF	[6,7,8,9,10,11,12,13,14,15,16,17,18,19,20,21,22]	[6,10,36,37,38,39,40,41]

^1^ (-) Electric charge was unstable or an electric breakdown took place.

## Data Availability

The data presented in this study are available on request from the corresponding author.

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
