# Peer review of "Corona Charging of Isotactic-Polypropylene Composites"

_polymers, 2021, doi:10.3390/polym13060942_

Round 1
Reviewer 1 Report
The paper presents a real contribution in the field of specific materials, and constitutes in fact a complement to a previous contribution. The core of the results is clear, but the initial description and the final discussion could be slightly reinforced. This will improve the comprehension and relevance of the contribution.
It could be published with some light modifications defined in pdf attached document. These additions are given to obtain a good balance in the discussion proposed in the paper.

Author Response
We would like to thank the Reviewers for the reviews, all the remarks have been taken into account. Below there are the answers.
Reviewer 1
Lines 28-88 (§1) The general presentation is convenient, and refers to a previous contribution, which is interesting for understanding the present one. The paragraph ends with a short description of the proposed objective.
However, it might be useful to expand this last paragraph with two or three additional sentences, while specifying - in a short additional paragraph – the issues at stake in these developments.
The answer, lines 89-97:
It is known that the polyolefin composite polarization with a high-intensity electric field allows producing stable electrets in which the injected electric charge lasts at least several months [14,15]. Corona discharges can be also a polarizing factor of hydrophobic polymers. However, the surface hydrophilicity of samples achieved by this method is reversible (or partially reversible), thus, the durability of the piezoelectric properties should be tested. From a practical point of view, however, it is important to produce a material exhibiting the permanent piezoelectric effect because such material can be used, for instance, in medicine for biomedical sensor production. Hence, a long-term stability study of piezoelectric properties is essential.
Lines 91-95 (§2.1)
The material properties refer to a previous work. It is useful here to give a brief summary of these properties, in the form of 3-4 additional lines.
The answer, lines 104-104:
The characteristics of the fillers have been added to this part.
Sillikolloid P87 contained 80 % of SiO2, 14 % of Al2O3, and less than 1 % of Fe2O3. The particle size was 6 μm - d97 and 15 μm - d50. The filler particles had the shape of round grains and lamellar aggregates.
Perlite PEX-02/20 was built of 65-75 % of SiO2, 10-18 % of Al2O3, 1-5 % of Fe2O3, and a few percent of alkali metal oxides alkaline earth metal oxides. The particle size was less than 20 μm – 60 %, 20-32 μm – 23.8 %, 32-40 μm – 5.4 %, 40-63 μm – 9.9 %, 63-100 μm – 0.6 %. The perlite particles looked like crushed bubbles.
Glass beads MinTron 7TM consisted of 48- 60% of SiO2, 20-30 % of Al2O3, 3-7 % of Fe2O3, and 5-9 % of alkali oxides The particle size was 20-30 μm - d90 and 5-9 μm - d50. The filler particles had regular spherical shapes.
A detailed description of the materials was included in previous works [14,15,21,22]
Lines 97-139 (§2.2-2.4) This description is relatively clear.
Nevertheless, each paragraph could be completed by 1-2 additional lines indicating how the chosen solution is well adapted to the problem.
The answers
at the end of 2.2. lines 148-150:
The apparatus presented in Figure 1 was designed and constructed to modify polymer films in the process of orientation and corona discharges in one technological line. The parameters of the device allow for the production of several meters long ribbons of a modified polymer 20 cm wide.
at the end of 2.3. lines 159-162:
The same method and conditions for testing the piezoelectric properties of samples were applied for the composites polarized with corona discharges and with an electric field. Maintaining the same measurement methodology is necessary to compare the properties of piezoelectrets created in two different ways.
at the end of 2.4. lines 171-173:
The AFM technique was chosen to study subtle changes in the surface topography of samples on a nanometric scale at high resolution, in particular, to observe the roughness profiles.
Lines 306-338 (§4) The discussion is interesting but a little short.
In particular, the second and third paragraphs (lines 315-334) could be reinforced in the sense of criticity / robustness. It is useful to add explanations on the general confidence given by the results. This can be made by a 3-4 lines additional paragraph.
The answer, lines 394-411:
The electrets made of polymer films of different compositions and morphology have different lifetimes. In the case of corona charging, the lifetime mainly depends on the number and energy of the charge traps on the film surface. Various defects in the chemical structure (e.g. branching of polymer chains or impurities from the polymerization process) and crystalline irregularities (e.g. edges, dislocations) are such traps.
It should be remembered that corona discharges can result in the formation of oxidized polymer structures (hydroxyl and carbonyl groups) which, however, are difficult to monitor with ATR-FTIR spectroscopy because probably these polar products can be placed in a thinner layer than could be detected by this method. The presence of these groups formed under the influence of corona discharges was confirmed by measurements of contact angles [21,27,28]. Such oxidized groups increase the polarity of the inherently hydrophobic polyolefins, which contributes to the increase in interactions with the charges induced in the activation process.
The piezoelectric charge could be permanently trapped in a specific place, but before it happens, the charge could move to the inner layers of the film due to the semi-crystalline structure of the polymer, which was usually described as a hopping mechanism [44,45]. In this way, the charge may be prevented from neutralization under the influence of external environmental factors, which results in the stabilization of piezoelectric properties.
- Hilczer, B.; Małecki, J. Electrets and piezopolymers, PWN: Warsaw, Poland, 1992; pp.15-22.
- Rychkov, D.; Altafim, R.A.P. Polymer Electrets and Ferroelectrets as EAPs: Models. In Electromechanically Active Polymers. Polymers and Polymeric Composites: A Reference Series, Carpi, F. Springer, Cham, 2016; pp. 945-959.
Reviewer 2 Report
The paper entitled " Corona Charging of Isotactic-Polypropylene Composites ( Jolanta Kowalonek , Halina Kaczmarek, BogusÅ‚aw Królikowski , Ewa Klimiec , and Marta ChyliÅ„ska) presents a new approach to obtaining piezoelectric polymeric films based on the isotactic-poly-12 propylene (i-PP) using corona discharge with the energy of 45 W·min/m2
I have listed several minor comments:
A. Some references must be included:
1. Vlaeva, I., Yovcheva, T., Viraneva, A., Kitova, S., Exner, G., Guzhova, A., & Galikhanov, M. (2012, December). Contact angle analysis of corona treated polypropylene films. In Journal of Physics: Conference Series (Vol. 398, No. 1, p. 012054). IOP Publishing.
2. Sellin, N., & Campos, J. S. D. C. (2003). Surface composition analysis of PP films treated by corona discharge. Materials Research, 6(2), 163-166.
3. Ramazanov, M. A., Hajiyeva, F. V., & Maharramov, A. M. (2016). Influence of corona discharge on the electret and charge states of nanocomposites based on isotactic polypropylene and zirconium dioxide nanoparticles. Ferroelectrics, 493(1), 103-109.
B. Infrared spectroscopy (FTIR/ATR) could be a valuable tool in order to investigate the change in the morphology and provided qualitative results of the chemical composition (new structures).
C. The changes in the surface free energy could be investigated by means of contact angle
measurements.
Author Response
We would like to thank the Reviewers for the reviews, all the remarks have been taken into account. Below there are the answers.
Reviewer 2:
- The suggested publications have been cited. The number of the cited positions in publication is in square brackets:
The sentence in lines 200-202 was added: “The corona discharges also increase the surface polarity, which was confirmed by contact angle measurements [21,27,28].”
- Vlaeva, I., Yovcheva, T., Viraneva, A., Kitova, S., Exner, G., Guzhova, A., & Galikhanov, M. (2012, December). Contact angle analysis of corona treated polypropylene films. J. Phys. Conf. Ser. (Vol. 398, No. 1, p. 012054). IOP Publishing. [27]
- Sellin, N., & Campos, J. S. D. C. (2003). Surface composition analysis of PP films treated by corona discharge. Mater. Res. 6(2), 163-166. [28]
- Ramazanov, M. A., Hajiyeva, F. V., & Maharramov, A. M. (2016). Influence of corona discharge on the electret and charge states of nanocomposites based on isotactic polypropylene and zirconium dioxide nanoparticles. Ferroelectrics, 493(1), 103-109 [26].
- ATR-FTIR spectra of the untreated i-PP composite films were presented in references [15,22]. Generally, the spectra of the neat PP and PP composites with the fillers were similar. Subtle differences were detected in the region 1000-1200 cm-1 for films with Sillikolloid and perlite. These differences resulted from the presence of the band of Si-O bond vibrations.
The corona treatment did not cause changes detected by ATR-FTIR spectroscopy. So, the infrared spectra were not included in the publication.
- The results of contact angle measurements and calculation of surface free energy of the untreated oriented and unoriented PP films and the composites are in ref. 22. All sample surfaces were very hydrophobic and again the differences between them were very small. However, the surface properties for these samples after corona treatment have not been investigated. Such data were collected only for HDPE and MDPE and the results showed a significant increase in surface polarity after corona treatment, especially for the samples with the fillers. As we had so many samples to study, not all research was successfully carried out. Now, we are not able to conduct the measurements because to produce a material we have to start the technological line and use a large amount of polymer and at this moment we are unable to do this. We can suspect based on literature and our previous results that the PP samples could behave similarly to PE samples (that is, the polarity of the samples increased significantly due to the action of corona discharge).